# Energy-Based Contrastive Learning of Visual Representations

**Beomsu Kim**
Department of Mathematical Sciences
KAIST
beomsu.kim@kaist.ac.kr

**Jong Chul Ye**
Kim Jaechul Graduate School of AI
KAIST
jong.ye@kaist.ac.kr

## Abstract

Contrastive learning is a method of learning visual representations by training Deep Neural Networks (DNNs) to increase the similarity between representations of positive pairs (transformations of the same image) and reduce the similarity between representations of negative pairs (transformations of different images). Here we explore Energy-Based Contrastive Learning (EBCLR) that leverages the power of generative learning by combining contrastive learning with Energy-Based Models (EBMs). EBCLR can be theoretically interpreted as learning the joint distribution of positive pairs, and it shows promising results on small and medium-scale datasets such as MNIST, Fashion-MNIST, CIFAR-10, and CIFAR-100. Specifically, we find EBCLR demonstrates from $\times 4$ up to $\times 20$ acceleration compared to SimCLR and MoCo v2 in terms of training epochs. Furthermore, in contrast to SimCLR, we observe EBCLR achieves nearly the same performance with 254 negative pairs (batch size 128) and 30 negative pairs (batch size 16) per positive pair, demonstrating the robustness of EBCLR to small numbers of negative pairs. Hence, EBCLR provides a novel avenue for improving contrastive learning methods that usually require large datasets with a significant number of negative pairs per iteration to achieve reasonable performance on downstream tasks. Code: https://github.com/1202kbs/EBCLR

## 1 Introduction

In computer vision, supervised learning requires a large-scale human-annotated dataset of images to train accurate deep neural networks (DNNs). However, acquiring labels for millions of images can be difficult or impossible in practice. This has led to the rise of self-supervised learning, which learns useful visual representations by forcing DNNs to be invariant or equivariant to image transformations. Among self-supervised learning algorithms, contrastive methods are rapidly gaining popularity for their superb performance.

Specifically, contrastive learning methods [1, 2, 3, 4, 5] train DNNs by increasing the similarity between representations of *positive pairs* (transformations of the same image) and decreasing the similarity between representations of *negative pairs* (transformations of different images). The negative pairs prevent DNNs from collapsing to the trivial constant function. There are numerous contrastive learning methods, such as SimCLR [4], Momentum Contrast (MoCo) [5], etc.

Despite this flurry of research in contrastive learning, contrastive methods require large datasets and a large number of negative pairs per positive pair to achieve reasonable performance on downstream tasks. Although there are recently proposed non-contrastive methods such as BYOL [6] and SimSiam [7] that do not rely on negative pairs, they require heuristic techniques such as stop-gradient to avert collapsing to the trivial solution. There has been an effort to explain the dynamics of non-contrastive methods with linear neural networks [3], but it is unclear how the analyses generalize to DNNs.

36th Conference on Neural Information Processing Systems (NeurIPS 2022).

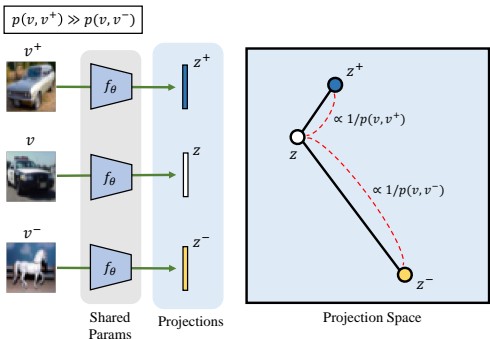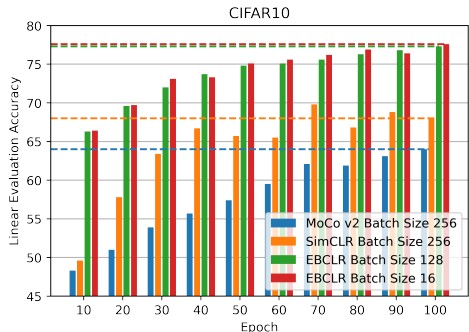

Figure 1: **Left:** An illustration of EBCLR. Here, $\propto$ means "is a monotonically increasing function of". We use $p(v, v')$, the joint distribution of positive pairs, as a measure of semantic similarity of images. Specifically, $p(v, v')$ will be high when $v$ and $v'$ are semantically similar, and low otherwise. A DNN $f_\theta$ is trained such that the distance in the projection space is controlled by $1/p(v, v')$. **Right:** Comparison of EBCLR, SimCLR, and MoCo v2 on CIFAR10 in terms of linear evaluation accuracy. EBCLR at epoch 10 beats MoCo v2 at epoch 100, and EBCLR at epoch 20 beats SimCLR and MoCo v2 at epoch 100. Moreover, EBCLR shows identical performance regardless of whether we use 254 negative pairs (batch size 128) or 30 negative pairs (batch size 16) per positive pair.

In this paper, we explore a novel avenue in visual representation learning: Energy-Based Contrastive Learning (EBCLR) which leverages the power of generative learning [8, 9, 10] by combining contrastive learning with energy-based models (EBMs). EBCLR complements the contrastive learning loss with a generative loss, and it can be interpreted as learning the joint distribution of positive pairs. In fact, we demonstrate that the existing contrastive loss is a special case of the EBCLR loss if the generative term is not used. Although EBMs are notorious for being difficult to train due to their reliance on Stochastic Gradient Langevin Dynamics (SGLD) [11], another important contribution of this work is that we overcome this by appropriate modifications to SGLD.

Extensive experiments on a variety of small and medium-scale datasets demonstrate that EBCLR is robust to small numbers of negative pairs, and it outperforms SimCLR and MoCo v2 [12] in terms of sample efficiency and linear evaluation accuracy. Hence, EBCLR opens up a new research direction for alleviating the dependence of contrastive methods on large datasets and large batches.

Our contributions can be summarized as follows:

- We propose a novel contrastive learning method called EBCLR which learns the joint distribution of positive pairs. We show that EBCLR loss is equivalent to a combination of a contrastive term and a generative term (Section 3). To the best of our knowledge, this is the first work to apply EBMs to contrastive learning of visual representations.

- We show that EBCLR offers two advantages over conventional contrastive learning methods: EBCLR is several times more sample efficient (Section 4.1) and robust to small batch sizes (Section 4.2). These factors lead to a non-trivial performance gain for EBCLR.

- We perform thorough ablation studies of the components of EBCLR: effect of changing the weight of the generative term (Section 4.3), effect of projection space dimension (Section 4.3), and the effect of the proposed SGLD modifications (Section 4.4).

## 2 Related Works

In this section, we go over related works necessary for understanding EBCLR. In Appendix A, we give a more extensive review of relevant works for those not familiar with EBMs, contrastive learning, or generative models.

### 2.1 Contrastive Learning

For a given batch of images $\{x_n\}_{n=1}^N$ and two image transformations $t$, $t'$, contrastive learning methods first create two views $v_n = t(x_n)$, $v'_n = t'(x_n)$ of each instance $x_n$. Here, the pair $(v_n, v'_m)$

is called a *positive pair* if $n = m$ and a *negative pair* if $n \neq m$. Given a DNN $f_\theta$, the views are then embedded into the projection space by passing the views through $f_\theta$ and normalizing.

Contrastive methods train $f_\theta$ to increase agreement between projections of positive pairs and decrease agreement between projections of negative pairs. Specifically, $f_\theta$ is trained to maximize the InfoNCE objective [1]. After training, outputs from the final layer or an intermediate layer of $f_\theta$ are used for downstream tasks.

There are numerous variants of contrastive methods. For instance, SimCLR [4] uses a composition of random cropping, random flipping, color jittering, color dropping, and blurring as the image transformation. Negative pairs are created by transforming different images within a batch. On the other hand, MoCo [12] maintains a queue of negative samples, so negative samples are not limited to views of images from the same batch.

## 2.2 Energy-Based Models

Given a scalar-valued *energy function* $E_\theta(v)$ with parameter $\theta$, an energy-based model (EBM) [13] defines a distribution by the formula

$$q_\theta(v) := \frac{1}{Z(\theta)} \exp\{-E_\theta(v)\} \tag{1}$$

where $Z(\theta)$ is the *partition function* which guarantees $q_\theta$ integrates to 1. Since there are essentially no restrictions on the choice of the energy function, EBMs have great flexibility in modeling distributions. Hence, EBMs have been applied to a wide variety of machine learning tasks, such as dimensionality reduction via autoencoding [14], learning generative classifiers [8, 9, 10, 15], generating images [16], and training regression models [17, 18]. Wang et al. [19] have explored connections between EBMs and InfoNCE to enhance generative performance of EBMs. However, to the best of our knowledge, this paper is the first to combine EBMs with contrastive learning for representation learning.

Given a target distribution, an EBM can be used to estimate its density $p$ when we can only sample from $p$. One way of achieving this is by minimizing the Kullback-Leibler (KL) divergence between $q_\theta$ and $p$ that maximizes the expected log-likelihood of $q_\theta$ under $p$ [20]:

$$\max_\theta \; \mathbb{E}_p[\log q_\theta(v)]. \tag{2}$$

Stochastic gradient ascent can be used to solve (2) [20]. Specifically, the gradient of the expected log-likelihood with respect to the parameters $\theta$ can be shown to be

$$\nabla_\theta \mathbb{E}_p[\log q_\theta(v)] = \mathbb{E}_{q_\theta}[\nabla_\theta E_\theta(v)] - \mathbb{E}_p[\nabla_\theta E_\theta(v)]. \tag{3}$$

Hence, updating $\theta$ with (3) amounts to pushing up on the energy for samples from $q_\theta$ and pushing down on the energy for samples from $p$. This optimization method is also known as contrastive divergence [21].

While the second term in (3) can be easily calculated as we have access to samples from $p$, the first term requires sampling from $q_\theta$. Previous works [16, 22, 10, 15] have used Stochastic Gradient Langevin Dynamics (SGLD) [11] to generate samples from $q_\theta$. Specifically, given a sample $v_0$ from some proposal distribution $q_0$, the iteration

$$v_{t+1} = v_t - \frac{\alpha_t}{2} \nabla_{v_t} E_\theta(v_t) + \epsilon_t, \quad \epsilon_t \sim \mathcal{N}(0, \sigma_t^2) \tag{4}$$

guarantees that the sequence $\{v_t\}$ converges to a sample from $q_\theta$ assuming $\{\alpha_t\}$ decays at a polynomial rate [11].

However, SGLD requires an infinite number of steps until samples from the proposal distribution converge to samples from the target distribution. This is unfeasible, so in practice, only a finite number of steps along with constant step size, i.e. $\alpha_t = \alpha$ and constant noise variance $\sigma_t = \sigma^2$ are used [16, 22, 10, 15]. Moreover, Yang and Ji [15] noted SGLD often generates samples with extreme pixel values that cause EBMs to diverge during training. Hence, they have proposed *proximal SGLD* which clamps gradient values into an interval $[-\delta, \delta]$ for a threshold $\delta > 0$. Then, the update equation becomes

$$v_{t+1} = v_t - \alpha \cdot \text{clamp}\{\nabla_v E_\theta(v_t), \delta\} + \epsilon \tag{5}$$

for $t = 0, \ldots, T - 1$, where $\epsilon \sim \mathcal{N}(0, \sigma^2)$ and $\text{clamp}\{\cdot, \delta\}$ clamps each element of the input vector into $[-\delta, \delta]$. In our work, we introduce additional modifications to SGLD which accelerate the convergence of EBCLR.

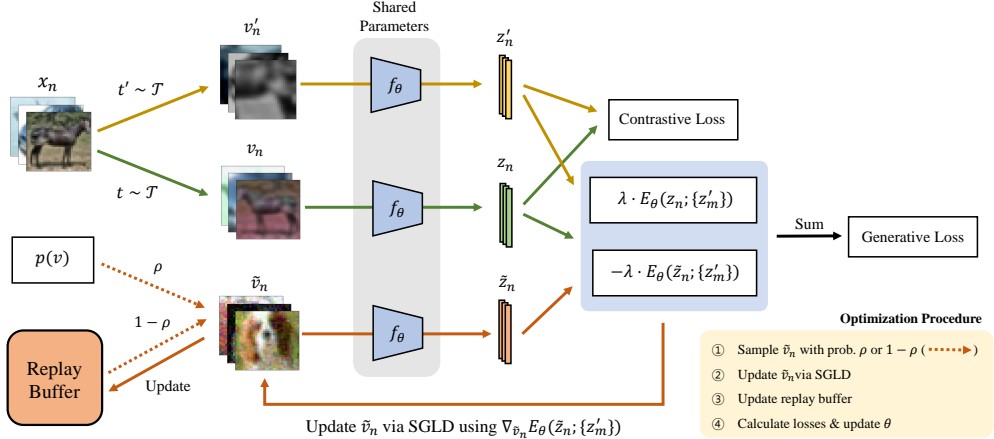

Figure 2: An illustration of the learning process of EBCLR.

## 3 Theory

### 3.1 Energy-Based Contrastive Learning

Let $\mathcal{D}$ be a distribution of images and $\mathcal{T}$ a distribution of stochastic image transformations. Given $x \sim \mathcal{D}$ and i.i.d. $t, t' \sim \mathcal{T}$, our goal is to approximate the joint distribution of the views

$$p(v, v'), \quad \text{where} \ \ v = t(x), \ v' = t'(x)$$

using the model distribution

$$q_\theta(v, v') \coloneqq \frac{1}{Z(\theta)} \exp\{-\|z - z'\|^2/\tau\}. \tag{6}$$

where $Z(\theta)$ is a normalization constant, $\tau > 0$ is a temperature hyper-parameter, and $z$ and $z'$ are projections computed by passing the views $v$ and $v'$ through the DNN $f_\theta$ and then normalizing to have unit norm. We now explain the intuitive meaning of matching $q_\theta$ to $p$.

Our key idea is to use $p(v, v')$ as a measure of semantic similarity of $v$ and $v'$. If two images $v$ and $v'$ are semantically similar, they are likely to be transformations of similar images. So, $p(v, v')$ will be high when $v$ and $v'$ are semantically similar and low otherwise. Suppose $q_\theta$ successfully approximates $p$. If we equate $p(v, v')$ to $q_\theta(v, v')$ in (6) and solve for $\|z - z'\|$, we see that the distance between $z$ and $z'$ will become a monotone increasing function of $1/p(v, v')$, which is the inverse of semantic similarity of $v$ and $v'$. So, semantically similar images will have nearby projections, and dissimilar images will have distant projections. This idea is illustrated in Figure 1.

To approximate $p$ using $q_\theta$, we train $f_\theta$ to maximize the expected log-likelihood of $q_\theta$ under $p$:

$$\max_\theta \ \mathbb{E}_p[\log q_\theta(v, v')]. \tag{7}$$

In order to solve this problem with stochastic gradient ascent, we could naively extend (3) to the setting of joint distributions to obtain the following result.

**Proposition 1.** *The the joint distribution* (6) *can be formulated as an EBM*

$$q_\theta(v, v') = \frac{1}{Z(\theta)} \exp\{-E_\theta(z, z')\}, \qquad E_\theta(v, v') = \|z - z'\|^2/\tau \tag{8}$$

*and the gradient of the objective of* (7) *is given by*

$$\nabla_\theta \mathbb{E}_p[\log q_\theta(v, v')] = \mathbb{E}_{q_\theta}[\nabla_\theta E_\theta(v, v')] - \mathbb{E}_p[\nabla_\theta E_\theta(v, v')]. \tag{9}$$

However, computing the first expectation in (9) requires sampling pairs of views $(v, v')$ from $q_\theta(v, v')$ via SGLD, which could be expensive. To avert this problem, we use Bayes' rule to decompose

$$\mathbb{E}_p[\log q_\theta(v, v')] = \mathbb{E}_p[\log q_\theta(v' \mid v)] + \mathbb{E}_p[\log q_\theta(v)] \quad \text{where} \quad q_\theta(v) = \int q_\theta(v, v') \, dv'. \tag{10}$$

In the first equation of (10), the first and second terms at the RHS will be referred to as discriminative and generative terms, respectively, throughout the paper. A similar decomposition was used by Grathwohl et al. [10] in the setting of learning generative classifiers.

Furthermore, we add a hyper-parameter $\lambda$ to balance the strength of the discriminative term and the generative term. The advantage of this modification will be discussed in Section 4.3. This yields our Energy-Based Contrastive Learning (EBCLR) objective

$$\mathcal{L}(\theta) := \mathbb{E}_p[\log q_\theta(v' \mid v)] + \lambda \mathbb{E}_p[\log q_\theta(v)]. \tag{11}$$

The discriminative term can be easily differentiated since the partition function $Z(\theta)$ cancels out when $q_\theta(v, v')$ is divided by $q_\theta(v)$. However, the generative term still contains $Z(\theta)$. We now present our key result, which is used to maximize (11). The proof is deferred to Appendix C.1.

**Theorem 2.** *The marginal distribution in* (10) *can be formulated as an EBM*

$$q_\theta(v) = \frac{1}{Z(\theta)} \exp\{-E_\theta(v)\}, \qquad E_\theta(v) := -\log \int e^{-\|z-z'\|^2/\tau} \, dv' \tag{12}$$

*where $Z(\theta)$ is the partition function in* (6)*, and the gradient of the generative term is given by*

$$\nabla_\theta \mathbb{E}_p[\log q_\theta(v)] = \mathbb{E}_{q_\theta(v)}[\nabla_\theta E_\theta(v)] - \mathbb{E}_p[\nabla_\theta E_\theta(v)]. \tag{13}$$

*Thus, the gradient of the EBCLR objective is*

$$\nabla_\theta \mathcal{L}(\theta) = \mathbb{E}_p[\nabla_\theta \log q_\theta(v' \mid v)] + \lambda \mathbb{E}_{q_\theta(v)}[\nabla_\theta E_\theta(v)] - \lambda \mathbb{E}_p[\nabla_\theta E_\theta(v)] \tag{14}$$

Theorem 2 suggests that the EBM for the joint distribution can be learned by computing the gradients of the discriminative term and the EBM for the marginal distribution. Moreover, we only need to sample $v$ from $q_\theta(v)$ to compute the second expectation in (14).

## 3.2 Approximating the EBCLR Objective

To implement EBCLR, we need to approximate expectations in (11) with their empirical means. Suppose samples $\{(v_n, v'_n)\}_{n=1}^N$ from $p(v, v')$ are given, and let $\{(z_n, z'_n)\}_{n=1}^N$ be the corresponding projections. As the learning goal is to make $q_\theta(v_n, v'_n)$ approximate the joint probability density function $p(v_n, v'_n)$, the empirical mean $\widehat{q}_\theta(v_n)$ can be defined as:

$$\widehat{q}_\theta(v_n) = \frac{1}{N'} \sum_{v'_m : v'_m \neq v_n} q_\theta(v_n, v'_m) \tag{15}$$

where the sum is over the collection of $v'_m$ defined as

$$\{v'_m : v'_m \neq v_n\} := \{v_k\}_{k=1}^N \cup \{v'_k\}_{k=1}^N - \{v_n\} \tag{16}$$

and $N' := |\{v'_m : v'_m \neq v_n\}| = 2N - 1$. One could also use a simpler form of the empirical mean:

$$\widehat{q}_\theta(v_n) = \frac{1}{N} \sum_{m=1}^N q_\theta(v_n, v'_m) \tag{17}$$

Similarly, $q_\theta(v'|v)$ in (11), which should approximate the conditional probability density $p(v'|v)$, can be represented in terms of $q_\theta(v_n, v'_n)$. Specifically, we have

$$q_\theta(v'_n \mid v_n) \simeq \frac{q_\theta(v_n, v'_n)}{\widehat{q}_\theta(v_n)} = \frac{q_\theta(v_n, v'_n)}{\frac{1}{N'} \sum_{v'_m : v'_m \neq v_n} q_\theta(v_n, v'_m)} = \frac{e^{-\|z_n - z'_n\|^2/\tau}}{\frac{1}{N'} \sum_{v'_m : v'_m \neq v_n} e^{-\|z_n - z'_m\|^2/\tau}} \tag{18}$$

It is then immediately apparent that the empirical form of the discriminative term using (18) is a particular instance of the contrastive learning objective such as InfoNCE and SimCLR. Hence, EBCLR can be interpreted as complementing contrastive learning with a generative term defined by an EBM. We will demonstrate in Section 4.1 that the generative term offers significant advantages over other contrastive learning methods.

For the second term, we use the simpler form of the empirical mean in (17):

$$\widehat{q_\theta}(v_n) = \frac{1}{N} \sum_{m=1}^{N} q_\theta(v_n, v'_m) = \frac{1}{Z(\theta)} \cdot \frac{1}{N} \sum_{m=1}^{N} \exp\{-\|z_n - z'_m\|^2/\tau\} \qquad (19)$$

We could also use (15) as the empirical mean, but either choice showed identical performance (see Appendix E.3). So, we have found (15) to be not worth the additional complexity, and have resorted to the simpler approximation (17) instead. In Appendix C.2, we theoretically justify that EBCLR will work as intended even with the approximations (15) or (17). If we compare (19) with (12), we can see that this approximation of $q_\theta(v)$ yields the energy function (after ignoring the constant $\log N$)

$$E_\theta(v; \{v'_m\}_{m=1}^N) := -\log\left( \sum_{m=1}^{N} e^{-\|z-z'_m\|^2/\tau} \right). \qquad (20)$$

### 3.3 Modifications to SGLD

According to Theorem 2, we need samples from the marginal $q_\theta(v)$ to calculate the second expectation in (14). Hence, we apply proximal SGLD (5) with the energy function (20) to sample from $q_\theta(v)$ as

$$\tilde{v}_{t+1} = \tilde{v}_t - \alpha \cdot \text{clamp}\{\nabla_v E_\theta(\tilde{v}_t; \{v'_m\}_{m=1}^N), \delta\} + \epsilon \qquad (21)$$

for $t = 0, \ldots, T - 1$, where $\epsilon \sim \mathcal{N}(0, \sigma^2)$. We make three additional modifications to proximal SGLD to expedite the training process. From here on, we will be referring to proximal SGLD in (5) when we say SGLD.

First, we initialize SGLD from generated samples from previous iterations, and with probability $\rho$, we reinitialize SGLD chains from samples from a proposal distribution $q_0$. This is achieved by keeping a replay buffer $\mathcal{B}$ of SGLD samples from previous iterations. This technique of maintaining a replay buffer has also been used in previous works and has proven to be crucial for stabilizing and accelerating the convergence of EBMs [16, 10, 15].

Second, the proposal distribution $q_0$ is set to be the data distribution $p(v)$. This choice differs from those of previous works [16, 10, 15] which have either used the uniform distribution or a mixture of Gaussians as the proposal distribution.

Finally, we use *multi-stage SGLD (MSGLD)*, which adaptively controls the magnitude of noise added in SGLD. For each sample $\tilde{v}$ in the replay buffer $\mathcal{B}$, we keep a count $\kappa_{\tilde{v}}$ of number of times it has been used as the initial point of SGLD. For samples with a low count, we use noise of high variance, and for samples with a high count, we use noise of low variance. Specifically, in (5), we set

$$\sigma = \sigma_{\min} + (\sigma_{\max} - \sigma_{\min}) \cdot [1 - \kappa_{\tilde{v}}/K]_+. \qquad (22)$$

where $[\cdot]_+ := \max\{0, \cdot\}$, $\sigma_{\max}^2$ and $\sigma_{\min}^2$ are the upper and lower bounds on the noise variance, respectively, and $K$ controls the decay rate of noise variance. The purpose of this technique is to facilitate quick exploration of the modes of $q_\theta$ and still guarantee SGLD generates samples with sufficiently low energy. The pseudocodes for MSGLD and EBCLR are given in Algorithms 1 and 2, respectively, in Appendix B, and the overall learning flow of EBCLR is described in Figure 2.

## 4 Experiments

We now describe the experimental settings. A complete description is deferred to Appendix D.

**Baseline methods and datasets.** The baseline methods are SimCLR, MoCo v2, SimSiam, and BYOL. The hyper-parameters are chosen closely following the original works [4, 12, 7, 6]. We use four datasets: MNIST [23], Fashion MNIST (FMNIST) [24], CIFAR10, and CIFAR100 [25].

**DNN architecture.** We decompose $f_\theta = \pi_\theta \circ \phi_\theta$ where $\phi_\theta$ is the encoder network and $\pi_\theta$ is the projection network. Rather than using the output of $f_\theta$ for downstream tasks, we follow previous works [4, 5, 1, 2, 3, 6, 7] and use the output of $\phi_\theta$ instead. In our experiments, we set $\phi_\theta$ to be a ResNet-18 [26] up to the global average pooling layer and $\pi_\theta$ to be a 2-layer MLP with output dimension 128. However, we remove batch normalization because batch normalization hurts SGLD [16]. We also replace ReLU with leaky ReLU to expedite the convergence of SGLD. For the baselines, we use settings proposed in the original works while keeping the backbone fixed to be ResNet-18.

**Evaluation.** We evaluate the representations by training a linear classifier on top of frozen $\phi_\theta$.

| Dataset | MNIST | | FMNIST | | CIFAR10 | | CIFAR100 | |
|---------|---------|-----------|---------|-----------|---------|-----------|---------|-----------|
| Statistic | Accuracy | Rel. Eff. | Accuracy | Rel. Eff. | Accuracy | Rel. Eff. | Accuracy | Rel. Eff. |
| SimSiam | 98.6 | 0.1 | 87.4 | 0.1 | 70.4 | 0.25 | 38.3 | 0.1 |
| BYOL | **99.3** | 0.4 | 89.0 | 0.2 | 70.9 | 0.25 | 41.7 | 0.2 |
| SimCLR | 99.0 | 0.1 | 88.5 | 0.15 | 68.0 | 0.15 | 43.1 | 0.25 |
| MoCo v2 | 98.1 | 0.05 | 87.8 | 0.1 | 64.0 | 0.1 | 38.2 | 0.1 |
| EBCLR | **99.3** | – | **90.1** | – | **77.3** | – | **49.1** | – |

Table 1: Linear evaluation accuracy and efficiency relative to EBCLR. Efficiency of a method relative to EBCLR is calculated by the following formula: (number of epochs used by EBCLR to reach the final accuracy of the method) / (total number of training epochs).

## 4.1 Comparison with Baselines

We use batch size 128 for EBCLR and batch size 256 for the baseline methods following Wang et. al [27] and train each method for 100 epochs. Table 1 shows the result of training each method for 100 epochs. Observe that EBCLR consistently outperforms all baseline methods in terms of linear evaluation accuracy. Moreover, relative efficiency indicates EBCLR is capable of achieving the same level of performance as the baseline methods with much fewer training epochs. Concretely, we observe at least $\times 4$ acceleration in terms of epochs compared to contrastive methods. Hence, EBCLR is a much more desirable choice than SimCLR or MoCo v2 for learn-

| Direction | M → FM | FM → M | C10 → C100 | C100 → C10 |
|-----------|--------|--------|------------|------------|
| SimSiam | 86.9 | 97.2 | 39.5 | 64.0 |
| BYOL | **87.3** | 97.8 | 42.3 | 70.2 |
| SimCLR | 86.9 | 97.4 | 39.9 | 67.3 |
| MoCo v2 | 85.3 | 97.1 | 36.2 | 62.9 |
| EBCLR | **87.4** | **98.5** | **46.9** | **72.4** |

Table 2: Comparison of transfer learning results in the linear evaluation setting. Left side of the arrow is the dataset than the encoder was pre-trained on, and right side of the arrow is the dataset that linear evaluation was performed on. We use the following abbreviations. **M** : MNIST, **FM** : FMNIST, **C10** : CIFAR10, **C100** : CIFAR100.

ing visual representations when we have a small number of training samples.

We also investigate the transfer learning performance of EBCLR. Table 2 compares the transfer learning accuracies. EBCLR always outperforms the baseline methods, and the performance gap is especially large on CIFAR10 and CIFAR100. This indicates EBCLR learns visual representations that generalize well across datasets. Repeating the above experiments with longer training or KNN classification led to similar conclusions (see Appendixes E.1 and E.4, respectively).

## 4.2 Effect of Reducing Negative Pairs

We compared the performances of EBCLR and SimCLR as we reduced the number of negative pairs per positive pair. For MoCo v2, the negative samples are provided by a queue updated by a momentum encoder. On the other hand, for EBCLR and SimCLR, negative samples come from the same batch as the positive pair. So, we did not have a way of fairly comparing EBCLR and SimCLR with MoCo v2. Hence, we excluded MoCo v2 from this experiment.

We note that, according to (18), given a batch of size $N$, we obtain $2N - 2$ negative pairs for each positive pair. SimCLR also has $2N - 2$ negative pairs for each positive pair. Hence, we can conveniently compare the sensitivity of EBCLR and SimCLR to the number of negative pairs by varying the batch size.

Table 3 shows the result of training each method for 100 epochs with batch sizes in $\{16, 64, 128\}$. We make three important observations. First, EBCLR consistently beats SimCLR in terms of linear evaluation accuracy for every batch size. Second, EBCLR is invariant to the choice of batch size. This contrasts with SimCLR whose performance degrades as batch size decreases. Consequently, EBCLR with batch size 16 beats SimCLR with batch size 128. Finally, as a byproduct of the second observation, the efficiency of EBCLR relative to SimCLR increases as batch size decreases. These properties make EBCLR suitable for situations where we cannot use large batch sizes, e.g., when we

| Dataset | MNIST | | | FMNIST | | | CIFAR10 | | | CIFAR100 | | |
|---|---|---|---|---|---|---|---|---|---|---|---|---|
| Batch Size | 16 | 64 | 128 | 16 | 64 | 128 | 16 | 64 | 128 | 16 | 64 | 128 |
| SimCLR | 98.7 | 99.1 | 99.1 | 87.1 | 88.0 | 88.2 | 65.2 | 67.6 | 69.0 | 36.9 | 39.1 | 43.0 |
| EBCLR | **99.4** | **99.3** | **99.3** | **89.6** | **90.4** | **90.1** | **77.6** | **78.2** | **77.3** | **48.8** | **49.8** | **49.1** |
| Rel. Eff. | 0.05 | 0.1 | 0.15 | 0.05 | 0.15 | 0.1 | 0.1 | 0.15 | 0.2 | 0.1 | 0.15 | 0.25 |

Table 3: Linear evaluation accuracies and efficiencies relative to EBCLR with various batch sizes. Efficiency of SimCLR relative to EBCLR is calculated by the following formula: (number of epochs used by EBCLR with the same batch size to reach the final accuracy of SimCLR) / (total number of training epochs).

have a small number of GPUs. Repeating the experiments with longer training or KNN classification again led to similar conclusions (see Appendixes E.2 and E.4, respectively).

### 4.3 Effect of $\lambda$ and Projection Dimension

We explored the effect of changing the hyper-parameter $\lambda$ which controls the importance of the generative term relative to the discriminative term (see Equation (11)). Figure 3a shows the performance of EBCLR with various values of $\lambda$ as training progresses. We observe that naively using $\lambda = 1.0$ leads to poor results. The performance peaks at $\lambda = 0.1$, and then degrades as we further decrease $\lambda$.

This result has two crucial implications. First, the generative term plays a non-trivial role in EBCLR. Second, we need to strike a right balance between the discriminative term and the generative term to achieve good performance on downstream tasks[1].

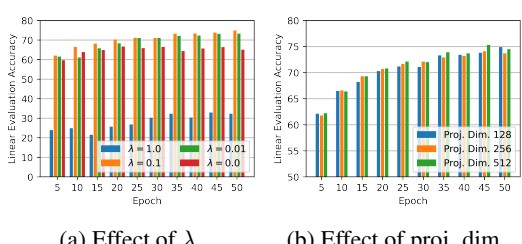

(a) Effect of $\lambda$.    (b) Effect of proj. dim.

Figure 3: Effect of $\lambda$ and projection dimension (output dimension of $\pi_\theta$, demonstrated on CIFAR10.

We also investigated the effect of varying the output dimension of $\pi_\theta$. Figure 3b shows linear evaluation results for projection dimensions in $\{128, 256, 512\}$. We observe that the projection dimension has essentially no influence on the training process. In this respect, EBCLR resembles SimCLR which is also invariant to the output dimension (see Figure 8 in the work by Chen et. al [4]).

### 4.4 Effect of SGLD Modifications

We now study the roles of the three SGLD modifications proposed in Section 3.3. Figure 4 shows the results of varying one parameter of MSGLD while keeping the others fixed.

**Effect of reinitialization frequency $\rho$.** Figure 4a displays linear evaluation results for $\rho \in \{0.0, 0.2, 1.0\}$. We note that setting $\rho = 1.0$ is equivalent to removing the replay buffer. Also, setting $\rho = 0.0$ is equivalent to never reinitializing SGLD chains.

Initially, $\rho = 0.0$ shows the best performance, as SGLD quickly reaches samples of lower energy. However, learning then slows down because of the lack of diversity of samples in the replay buffer $\mathcal{B}$. This implies that it is necessary to set $\rho > 0$ in order to learn good representations.

On the other hand, $\rho = 1.0$ shows slow convergence in the beginning because samples in the replay buffer are not given enough iterations to reach low energy. Although it does beat $\rho = 0.0$ at latter epochs, it still often performs worse than $\rho = 0.2$. Moreover, it is not sample-efficient compared to $\rho = 0.2$ since we have to provide an entire batch of new samples for reinitializing SGLD chains at each iteration.

---

[1]Interestingly, we observed a similar phenomenon when we used models trained with EBCLR to generate images. For more details, we refer the readers to Appendix E.5.

Given the above observations, it is clear why the intermediate value 0.2 is the best choice out of $\rho \in \{0.0, 0.2, 1.0\}$. $\rho = 0.2$ allows enough time for samples in the replay buffer to reach low energy while still maintaining the diversity of samples in $\mathcal{B}$. Also, it is sample-efficient compared to $\rho = 1.0$.

**Effect of proposal distribution** $q_0$. Figure 4b compares linear evaluation accuracies with $q_0$ as the uniform distribution and $q_0 = p(v)$. We observe prominent acceleration in the initial epochs for $q_0 = p(v)$. Hence, we can conclude that this choice of proposal distribution is crucial for the high efficiency of EBCLR compared to the baseline methods in Tables 1 and 3.

We believe this acceleration effect can be explained by the work of Hinton [21]. Specifically, let us observe that the EBM update equation (3) pushes up the energy on the model distribution $q_\theta$. In the implementation of EBCLR with $q_0 = p(v)$, however, $q_\theta$ is replaced by the distribution of samples created by a finite number of (noisy) gradient steps on real data points (see Section 3.3). Hence, the modified EMB update equation contains the curvature information of the data manifold. This curvature information may expedite the training process of EBCLR. For a detailed discussion on this, we refer the readers to Section 3 of the work by Hinton [21].

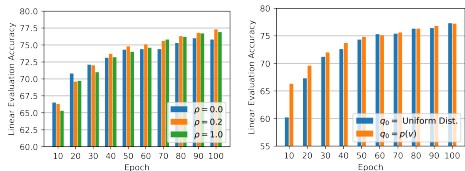
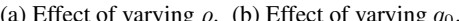

(a) Effect of varying $\rho$. (b) Effect of varying $q_0$.

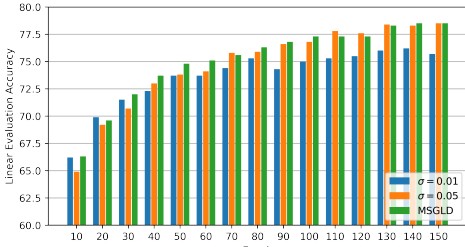

**Comparison of SGLD and MSGLD.** Figure 4c shows results with SGLD with $\sigma \in \{0.01, 0.05\}$ and MSGLD with $\sigma_{\min} = 0.01$ and $\sigma_{\max} = 0.05$. We note that setting $\sigma_{\min} = \sigma_{\max}$ reduces MSGLD to SGLD. We observe $\sigma = 0.01$ initially shows fast convergence but then saturates due to the lack of diversity of generated samples. On the other hand, $\sigma = 0.05$ initially has the worst performance but eventually beats $\sigma = 0.01$ since $\sigma = 0.05$ quickly explores the modes of $q_\theta$. MSGLD inherits the best of both settings. Specifically, MSGLD is as fast as $\sigma = 0.01$ in the beginning, and it does not suffer from the saturation problem.

(c) SGLD with $\sigma \in \{0.01, 0.05\}$ and MSGLD.

Figure 4: Ablation study of SGLD modifications on CIFAR10.

## 5 Limitations and Societal Impacts

**Limitations.** The main limitation of our work is of scale. While EBCLR demonstrates superior sample efficiency, it requires inner SGLD iterations (which cannot be parallelized) and a replay buffer $\mathcal{B}$. These two components increase the computational burden of EBCLR. So, we found it difficult to apply EBCLR to large-scale data such as ImageNet. However, we note that inner SGLD iterations and the replay buffer are not particular limitations of EBCLR, but limitations of EBMs in general. Given the increasing efforts to overcome these limitations such as Proximal-YOPO-SGLD (for more discussion, see Appendix F), we believe EBCLR will eventually be applicable to larger data.

**Social Impacts.** We generally expect positive outcomes from this research. Further development of EBCLR can mitigate the need for large amount of data and large batch sizes to learn good representations and ultimately lead to a reduction in resource consumption.

## 6 Conclusion

In this work, we proposed EBCLR which combines contrastive learning with EBMs. This amalgamation of ideas has led to both theoretical and practical contributions. Theoretically, EBCLR associates distance in the projection space with the density of positive samples. Since the distribution of positive samples reflects the semantic similarity of images, EBCLR is capable of learning good visual representations. Practically, EBCLR is several times more sample-efficient than conventional contrastive and non-contrastive learning approaches and is robust to small numbers of negative pairs. Hence, EBCLR is applicable even in scenarios with limited data or devices. We believe that EBCLR makes representation learning available to a wider range of machine learning practitioners.

## Acknowledgments and Disclosure of Funding

This work was supported by the National Research Foundation of Korea under Grant NRF-2020R1A2B5B03001980, KAIST Key Research Institute (Interdisciplinary Research Group) Project, and Field-oriented Technology Development Project for Customs Administration through National Research Foundation of Korea(NRF) funded by the Ministry of Science & ICT and Korea Customs Service(\*\*NRF-2021M3I1A1097938\*\*).

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
