# OpenReview forum: "Energy-Based Contrastive Learning of Visual Representations"
_NeurIPS.cc/2022/Conference — NeurIPS 2022 Accept_

### Official Review · Reviewer_oTyr · 2022-07-09

**Rating:** 7
**Confidence:** 4
**Soundness:** 3 good
**Presentation:** 3 good
**Contribution:** 4 excellent

**Summary:**

Summary:


The paper proposes an energy based contrastive learning framework that outperforms the existing contrastive learning methods such as SimCLR, MoCo, BYOL, SimSiam on four datasets, MNIST, FMNIST, CIFAR10, CIFAR100, using linear evaluations. In essence, the energy-based contrastive learning adds an energy-based generative term to the existing contrastive learning objective. Subsequently, the paper focusses on proposing relevant challenges to Stochastic Gradient Langevin Dynamics (SGLD) for improved representation learning. Overall, the paper performs some relevant experiments along with the ablation studies that indicate the usefulness of the EBCLR framework. However, the mathematical formulation has some obvious mistakes that the authors would need to clarify. Additionally, the paper is not well-written and requires proper narrative to better pitch the idea along with additional experiments.


Strengths:

- It was interesting to see that EBCLR boils down to standard contrastive learning when solved for the KL divergence between the joint distribution of the positive pairs modeled via EBM.
- The results of the proposed framework showcase that EBCLR outperforms all the existing methods on four datasets. Additionally, the authors show that EBCLR is more sample efficient as it achieves good performance with smaller batch sizes.
- The authors study the various aspects of the framework through ablation studies.

Weaknesses and Suggestions:

- The abstract and introduction present this work with a motivation to reduce the batch size used for improved representation learning through contrastive methods. However, it seems that being able to perform well with small batch size is a byproduct of EBCLR framework. I don’t see any connection between the formulations discussed in the paper towards making EBCLR sample-efficient. Unless the authors present relevant works which have shown EBMs to be sample-efficient or prove it themselves, I don’t buy their motivation directly.
- My another major concern is with the core observation itself. Years of previous works has shown that more data should improve the performance more. A good representation learning method should be able to perform better when encountered with more data. Line 235 claims that EBCLR is invariant to batch size which to me is a double edged sword — good performance with low batch size is a good thing but not improving much with more batch size is not good.
- Equation 20 is wrong. It is calculated by solving Eq 12 and Eq 19. The authors cancel out the normalization terms from Eq 12 and Eq 19 but they are not actually equal!
- Most of standard contrastive learning methods benchmark themselves on ImageNet dataset which paper does not do. The authors may use the latest FFCV library to fasten their ImageNet training pipeline.
- The paper uses linear probing accuracy to evaluate the model accuracies. To further understand the representation quality, the authors should perform K-NN classification in the representation space too.
- Further insights or discussions as to why EBCLR is more sample-efficient are definitely required.


Edits:
- Figure 1 left is not very clear. log 1/ p(v, v’) threw me off at the first place. You may need to explain it more.
- Eq.4 should have N(0, \sigma)
- Line 193 has repetitions

**Questions:**

Mentioned in the main review

**Ethics Review Area:**

["I don’t know"]

**Limitations:**

Mentioned in the main review

**Strengths And Weaknesses:**

Mentioned in the main review

---

> ### Author Response · Authors · 2022-08-01
> **Response to Reviewer oTyr (Part 4/4)**
>
> **Q5. The paper uses linear probing accuracy to evaluate the model accuracies. To further understand the representation quality, the authors should perform K-NN classification in the representation space too.**
>
> A. The following content has been added to Appendix E.4 of our revised paper.
>
> We reproduced Tables 1, 2, and 3 in our paper with K-NN classification in place of linear probing with the same weight checkpoints for all methods. We followed the protocol of [23] and used weighted K-NN classifier with K=20. The results are given below. In Review Table 3, except for the case of BYOL on MNIST, we see EBCLR beats all baselines again by a non-trivial margin. EBCLR is also shown to be sample efficient. In Review Table 4, we see EBCLR outperforms all baselines on the task of transfer learning. In Review Table 5, we see EBCLR is robust to small batch sizes. This leads to the lower efficiency of SimCLR compared to EBCLR. On CIFAR10 and CIFAR100, relative efficiency does not decrease below 0.05 since we saved checkpoints every 5 epochs. We can reasonably expect SimCLR to have lower relative efficiency at smaller batch sizes if we had saved checkpoints every epoch.
>
> *Review Table 3: KNN evaluation accuracy and efficiency relative to EBCLR.*
> | Dataset | MNIST |  | FMNIST |  | CIFAR10 |  | CIFAR100 |  |
> | --- | --- | --- | --- | --- | --- | --- | --- | --- |
> | Statistic | Accuracy | Rel. Eff. | Accuracy | Rel. Eff. | Accuracy | Rel. Eff. | Accuracy | Rel. Eff. |
> | SimSiam | 94.5 | 0.05 | 80.4 | 0.05 | 59.6 | 0.1 | 23.9 | 0.05 |
> | BYOL | **98.8** | - | 84.4 | 0.2 | 62.3 | 0.15 | 31.3 | 0.1 |
> | SimCLR | 97.4 | 0.15 | 84.1 | 0.15 | 57.2 | 0.05 | 29.5 | 0.1 |
> | MoCo v2 | 94.5 | 0.05 | 83.1 | 0.1 | 54.1 | 0.05 | 26.0 | 0.05 |
> | EBCLR | 98.1 | - | **86.6** | - | **71.4** | - | **39.4** | - |
>
> *Review Table 4: Comparison of transfer learning results in the KNN evaluation setting.*
> | Dataset | MNIST | FMNIST | CIFAR10 | CIFAR100|
> | --- | --- | --- | --- | --- |
> | SimSiam | 80.3 | 85.1 | 26.1 | 52.6 |
> | BYOL | 82.6 | 92.4 | 29.3 | 63.7 |
> | SimCLR | 80.7 | 87.6 | 25.4 | 56.0 |
> | MoCo v2 | 78.7 | 89.7 | 23.8 | 52.2 |
> | EBCLR | **83.4** | **95.5** | **35.7** | **65.4** |
>
> *Review Table 5: KNN evaluation accuracies and efficiencies relative to EBCLR with various batch sizes.*
> | Dataset | MNIST | | | FMNIST | | | CIFAR10 | | | CIFAR100 | | |
> | --- | --- | --- | --- | --- | --- | --- | --- | --- | --- | --- | --- | --- |
> | Batch Size | 16 | 64 | 128 | 16 | 64 | 128 | 16 | 64 | 128 | 16 | 64 | 128 |
> | SimCLR | 97.1 | 97.7 | 97.4 | 82.3 | 83.2 | 83.3 | 52.8 | 56.0 | 56.8 | 22.1 | 22.7 | 28.8 |
> | EBCLR | **98.5** | **98.3** | **98.1** | **85.0** | **85.9** | **86.6** | **72. 2** | **72.6** | **71.4** | **40.6** | **41.4** | **39.4** |
> | Rel. Eff. | 0.05 | 0.05 | 0.15 | 0.05 | 0.15 | 0.1 | 0.05 | 0.05 | 0.05 | 0.05 | 0.05 | 0.05 |
>
> **Q6. Figure 1 left is not very clear. log 1/ p(v,v’) threw me off at the first place. You may need to explain it more.**
>
> A. As shown in the revised version of our paper, we improved the clarity of Figure 1 by simplifying the notation and adding more explanation. We added more explanation in Section 3.1 as well. Please let us know if Figure 1 is still unclear. We will be happy to fix it.
>
> **Q7. Eq.4 should have N(0, \sigma)**
>
> A. We were thinking of standard Langevin dynamics, not SGLD, when we wrote Eq. (4). We fixed the typo.
>
> **Q8. Line 193 has repetitions**
>
> A. We have fixed the repetitions.
>
> **References**
>
> [23] Emerging Properties in Self-Supervised Vision Transformers, ICCV, 2021.

---

> > ### Comment · Reviewer_oTyr · 2022-08-07
> > **Response to Author Rebuttal**
> >
> > I would like to thank the reviewers for their diligent efforts towards the rebuttals. I have changed my score to 7 based on the author rebuttals.
> >
> > > **Clarification on EBCLR's sample efficiency.**
> >
> > I thank the authors for their insightful rebuttal. Adding A.4 in the paper has definitely made it more complete and sensible.
> >
> > > **Clarification on improvement with more data.**
> >
> > I thank the authors for their clarifications on this front, I understand their arguments. By more data, I meant more negative samples per positive pair rather than more data samples in the training set itself. I apologize for ill-positioning of my original review question.
> >
> > > **Clarification on the equations.**
> >
> > Got it, thank you!
> >
> > > **Results on the ImageNet dataset.**
> >
> > I completely understand. Please see if you can add your results on ImageNet too. It will make your paper more stronger in terms of empirical experiments. SimCLR, BYOL, MOCO V2 and SimSiam (and other contrastive learning papers) always benchmark their models on ImageNet. To understand the importance of benchmarking on ImageNet, you can read the abstracts of each of these papers to see how they talk about doing well on ImageNet specifically.
> >
> > Secondly, the authors might find it useful to check [1] which I came across recently that talks about labeling errors in the CIFAR-10,CIFAR-100 and ImageNet dataset.
> >
> > > **Results on k-NN**
> >
> > I thank the authors for presenting the results on K-NN in such a short period of rebuttal. I am impressed by the results; thanks!
> >
> > Reference:
> >
> > [1] Pervasive Label Errors in Test Sets Destabilize Machine Learning Benchmarks: https://arxiv.org/abs/2103.14749

---

> > > ### Author Response · Authors · 2022-08-07
> > > **Thank you!**
> > >
> > > Thank you for raising the score! We are certain your insights and the discussion have improved our work. We will also try our best to incorporate all suggestions into the next version of our paper.

---

> ### Author Response · Authors · 2022-08-01
> **Response to Reviewer oTyr (Part 3/4)**
>
> **Q3. Equation 20 is wrong. It is calculated by solving Eq 12 and Eq 19. The authors cancel out the normalization terms from Eq 12 and Eq 19 but they are not actually equal!**
>
> A. We inform the Reviewer that Eq. (12) and Eq. (19) are not equal in general, because Eq. (19) (which comes from from Eq. (17)) is a Monte-Carlo **approximation** of the marginal distribution in Eq. (12). In a high dimensional space, exact integration of a general function (in this case, obtaining a marginal density function from a joint density function) using Riemann sum is impossible, so it is a widely accepted practice to use Monte-Carlo approximations [13].
>
> We would also like to assure the reviewer that optimizing the EBCLR objective Eq. (11) with the approximations of Section 3.2 (and $\lambda = 1$) will still cause $q_\theta(v,v′)$ to approximate $p(v,v′)$, and thus achieve the goal of EBCLR. The following content has been added to Appendix C.2 of our revised paper.
>
> Specifically, it is known that optimizing the first term of Eq. (11) with Eq. (18) in place of $q_\theta(v’ \mid v)$ will cause Eq. (18) to approximate $p(v’ \mid v)$ [14]. Also, optimizing the second term of Eq. (11) with Eq. (15) or (17) in place of $q_\theta(v)$ will cause Eq. (15) or (17) to be proportional to $p(v)$ (indeed, in Appendix E.5 of the revised paper, we see SGLD samples of the EBCLR marginal $q_\theta(v)$ approximated by Eq. (17) achieve a non-trivial FID score). Thus, the product of Eq. (15) or (17) with Eq. (18) will be proportional to $p(v,v’)$. Moreover, by construction, the product of Eq. (15) or (17) with Eq. (18) is (approximately) $q_\theta(v,v’)$. Hence, optimizing the EBCLR objective Eq. (11) will cause $q_\theta(v,v’)$ to model $p(v,v’)$.
>
> For convenience of the Reviewer, we rewrite the relevant equations below.
>
> Eq. (11): $\mathcal{L}(\theta) \coloneqq \mathbf{E}_p[\log q_\theta(v’ \mid v)] + \lambda \mathbf{E}_p[\log q_\theta(v)]$
>
> Eq. (12): $q_\theta(v) = \frac{1}{Z(\theta)} \exp(-E_\theta(v)), \qquad E_\theta(v) \coloneqq - \log \int e^{-\|\|z- z'\|\|^2 / \tau} \, dv'$
>
> Eq. (15): $\widehat q_\theta(v_n) = \frac{1}{N'} \sum_{v_m' : v'_m \neq v_n} q_\theta(v_n, v'_m)$
>
> Eq. (17): $\widehat q_\theta(v_n) = \frac{1}{N} \sum_{m = 1}^N q_\theta(v_n, v'_m)$
>
> Eq. (18): $q_\theta(v_n' \mid v_n) \simeq \frac{q_\theta(v_n, v'_n)}{\widehat q_\theta(v_n)} = \frac{q_\theta(v_n,v'_n)}{\frac{1}{N'} \sum q_\theta(v_n,v'_m)} = \frac{e^{-\|\| z_n - z'_n \|\|^2 / \tau}}{\frac{1}{N'} \sum e^{-\|\| z_n - z'_m \|\|^2 / \tau}}$ where the sums are over $v_m' : v'_m\neq v_n$
>
> Eq. (19): $\widehat q_\theta(v_n) = \frac{1}{N} \sum_{m = 1}^N q_\theta(v_n,v'_m) = \frac{1}{Z(\theta)} \cdot \frac{1}{N} \sum \exp(- \|\| z_n - z'_m \|\|^2 / \tau)$ where the last sum is from $m = 1$ to $N$.
>
> **Q4. Most of standard contrastive learning methods benchmark themselves on ImageNet dataset which this paper does not do. The authors may use the latest FFCV library to fasten their ImageNet training pipeline.**
>
> A. Thank you for the great suggestion. We tried to run EBCLR on ImageNet. But (due to reasons written in the limitations section / Section 5 of this paper) we were unable to produce results within the rebuttal period as we had to run hyperparameter searches for the baselines and EBCLR. We additionally note that many published works on EBMs [15-18] and adversarial training [19-22] (which also requires inner iterations to generate adversarial examples every model update step) train models only up to medium-scale datasets such as CIFAR-10/100. If we obtain ImageNet results, we will make sure to update our paper.
>
> **References**
>
> [13] An Introduction to MCMC for Machine Learning, Machine Learning, 2003.
>
> [14] Understanding hard negatives in noise contrastive estimation, NAACL, 2021.
>
> [15] Your classifier is secretly and energy-based model and you should treat it like one, ICLR, 2020.
>
> [16] Learning the Stein Discrepancy for Training and Evaluating Energy-Based Models without Sampling, ICML, 2020
>
> [17] Flow Contrastive Estimation of Energy-Based Models, CVPR, 2020.
>
> [18] JEM++: Improved Techniques for Training JEM, ICCV, 2021.
>
> [19] Theoretically Principled Trade-off between Robustness and Accuracy, ICML, 2019.
>
> [20] Unlabeled Data Improves Adversarial Robustness, NeurIPS, 2019.
>
> [21] Improving Adversarial Robustness Requires Revisiting Misclassified Examples, ICLR, 2020.
>
> [22] Learning Adversarially Robust Representations via Worst-Case Mutual Information Maximization, ICML, 2020.

---

> ### Author Response · Authors · 2022-08-01
> **Response to Reviewer oTyr (Part 2/4)**
>
> **Q2. Another major concern is with the core observation itself. Years of previous works has shown that more data should improve the performance more. A good representation learning method should be able to perform better when encountered with more data. Line 235 claims that EBCLR is invariant to batch size which to me is a double-edged sword — good performance with low batch size is a good thing but not improving much with more batch size is not good.**
>
> A. We would like to gently remind the Reviewer that the amount of data (number of data samples) encountered by a model during training increases proportional to the number of training epochs, not to batch size. This is because, regardless of the batch size, the model makes one sweep over the entire training dataset in a single epoch. So, all methods are compared after a fixed number of training epochs, not after a fixed number of model parameter updates.
>
> We also inform the Reviewer that recent works on representation learning view dependence on batch size as an undesirable behavior rather than an advantage, i.e., a good representation learning method should be robust to batch size [9,10,11,12]. The rationale is that a good representation learning method should reach optimal performance capable by a model rapidly regardless of batch size used. We believe EBCLR takes a step towards this goal, since EBCLR outperforms all baselines by a non-trivial margin regardless of batch size.
>
> Having this in mind, we indeed find the performance of EBCLR increases consistently when encountered with more data (larger number of training epochs). For instance, if we compare Table 1 and Table 4 (Appendix E.1), we see EBCLR at epoch 200 outperforms EBCLR at epoch 100. We summarize the relevant statistics in the table below. We also reproduce the same table with K-NN classification and find similar conclusion holds.
>
> *Review Table 1: EBCLR linear evaluation performance at epochs 100 and 200.*
> | Dataset | MNIST | FMNIST | CIFAR10 | CIFAR100|
> | --- | --- | --- | --- | --- |
> | Epoch 100 | 99.3 | 90.1 | 77.3 | 49.1 |
> | Epoch 200 | 99.4 | 90.8 | 80.0 | 51.6 |
>
> *Review Table 2: EBCLR KNN evaluation performance at epochs 100 and 200.*
> | Dataset | MNIST | FMNIST | CIFAR10 | CIFAR100|
> | --- | --- | --- | --- | --- |
> | Epoch 100 | 98.1 | 86.6 | 71.4 | 39.4 |
> | Epoch 200 | 98.2 | 86.9 | 74.4 | 43.3 |
>
> **References**
>
> [9] Bootstrap Your Own Latent, NeurIPS, 2020.
>
> [10] Improved Baselines with Momentum Contrastive Learning, arXiv, 2020.
>
> [11] Barlow Twins: Self-Supervised Learning via Redundancy Reuction, ICML, 2021.
>
> [12] VICReg: Variance-Invariance-Covariance Regularization for Self-Supervised Learning, ICLR, 2022.

---

> ### Author Response · Authors · 2022-08-01
> **Response to Reviewer oTyr (Part 1/4)**
>
> **Q1. The abstract and introduction present this work with a motivation to reduce the batch size used for improved representation learning through contrastive methods. However, it seems that being able to perform well with small batch size is a byproduct of EBCLR framework. I don’t see any connection between the formulations discussed in the paper towards making EBCLR sample-efficient. Unless the authors present relevant works which have shown EBMs to be sample-efficient or prove it themselves, I don’t buy their motivation directly. / Further insights or discussions as to why EBCLR is more sample-efficient are definitely required.**
>
> A. We acknowledge that sample efficiency and robustness to small batch size is more of a byproduct of our framework. Hence, we have removed or reworded possibly misleading sentences in the Abstract and the Introduction. However, as written below, we inform the Reviewer that there are many works which have shown EBMs (or generative models in general) to be sample-efficient. The following content has been reflected in the Abstract and the Introduction and added to Appendix A.4 of the revised paper.
>
> We motivate the sample efficiency of EBCLR through the lens of discriminative and generative learning. We note that standard contrastive learning (e.g., SimCLR) optimizes a discriminative objective (the model must predict positive pairs) and EBCLR optimizes a generative objective (given a pair of images, the model must output how likely it is to be a positive pair). So, the Reviewer’s concern boils down to comparing the performance of discriminative models and generative models.
>
> Having this in mind, we observe that there is a long line of works which show (theoretically and / or empirically) that generative models can outperform discriminative ones. For instance, [1] has theoretically shown that Normal Discriminant Analysis can be more efficient than logistic regression. [2] has also theoretically shown that in the small sample regime, naïve Bayes can be more resistant to overfitting than logistic regression (i.e., find better solutions when given small number of training samples).
>
> There are also works [3,4,5] that consider optimizing a weighted combination of a discriminative loss and a generative loss, just like our EBCLR objective Eq. (11), in a supervised learning setting. They find that with an appropriate weight, the model can outperform discriminative models. In particular, [3] finds this hybrid approach to be beneficial in the limited training data setting. [5] uses this hybrid approach to train a classifier, and finds that the classifier exhibits excellent calibration, unlike other calibration methods which require additional training data.
>
> We can also address the Reviewer’s question from the perspective of generative pre-training. [6] finds that pre-training a network with a generative loss brings the network weights towards a good local minimum, thus bringing better generalization. [7] finds that generative pre-training acts as a regularizer, and that this regularization effect can be beneficial when we have a small training set.
>
> Another explanation specific to EBCLR is that the generative loss has an effect orthogonal to the effect of the discriminative loss. To minimize the generative loss, the model needs to place low energy on data and high energy elsewhere. This can be achieved only when the model has features which accurately capture the manifold of the data. When we have a small training set, the model may easily solve the discriminative objective and overfit. By providing an auxiliary generative task, the overfitting can be mitigated. An analogous effect with Joint Energy-Based Models is demonstrated in [8] (for instance, Figure 1 in [8]).
>
> **References**
>
> [1] The Efficiency of Logistic Regression Compared to Normal Discriminant Analysis, JASA, 1975.
>
> [2] On Discriminative vs. Generative classifiers: A comparison of logistic regression and naïve Bayes, NIPS, 2001.
>
> [3] Principled Hybrids of Generative and Discriminative Models, CVPR, 2006.
>
> [4] Classification using Discriminative Restricted Boltzmann Machines, ICML, 2008.
>
> [5] Your Classifier is Secretly an Energy Based Model and You Should Treat it Like One, ICLR, 2020.
>
> [6] Greedy Layer-Wise Training of Deep Networks, NIPS, 2006
>
> [7] Why Does Unsupervised Pre-training Help Deep Learning?, JMLR, 2010.
>
> [8] Joint Energy-Based Models for Semi-Supervised Classification, ICML Workshop, 2020.

---

> ### Author Response · Authors · 2022-08-07
> **Does our response answer your concerns and questions?**
>
> Dear Reviewer oTyr,
>
> As the end of the author-reviewer discussion period is approaching, we would like to ask whether our response has addressed your concerns and questions adequately. If not, we would be happy to discuss further.
>
> Regards,
>
> the authors.

---

### Official Review · Reviewer_VTiU · 2022-07-11

**Rating:** 6
**Confidence:** 5
**Soundness:** 2 fair
**Presentation:** 3 good
**Contribution:** 3 good

**Summary:**

The paper proposes a new perspective to leverage contrastive self-supervised learning by combining energy-based models (EBMs). The code is to approximate the joint distribution of a pair of positive samples in contrastive learning by EBMs. Following the idea, the authors build a new contrastive learning method called EBCLR. To support the proposed method, the authors provide both theoretical and empirical evidence.

**Questions:**

I have put some of my concerns in the analysis of "Weaknesses" above. The clarification or improvements over them would be helpful for me to better justify the significance of this paper and to make my final rating more fairly and confident.

**Limitations:**

The authors have made the limitations analysis in the draft. I don't recognize the additional potential negative societal impact of this work.

**Strengths And Weaknesses:**

Strengths:
1. The authors provide both theoretical and empirical evidence to support their claim.
2. The experiment design is good, especially how it provides support for the efficiency of the proposed method given the limited number of negative pairs. Instead of the absolute performance advantage shown in Table 1, which is relatively not significant, the results in Table 3 are more impressive.
3. The paper is well written and easy to follow. It is well organized in most parts.
4. I would put it as a strength the limitation analysis by the authors which answers some of my concerns.

Weaknesses:
1. I have some concerns about the idea of connecting EMBs and contrastive learning has been studied for long, even not necessarily under the framework created by recent deep-contrastive-self-supervised learning, such as MoCo and SimCLR. Some related works might be missing for the overview of this area, such as:
* A Unified Contrastive Energy-based Model for Understanding the Generative Ability of Adversarial Training, ICLR 2022
* A Unified Energy-Based Framework for Unsupervised Learning, Artificial Intelligence and Statistics. PMLR, 2007.
2. Some parts of the paper is not well organized. For example, at the end of the Introduction section, the authors claim five Contributions, but not all of them are well organized in the experiment section and following discussions. Besides this, it would be not convincing to put some minor points about of the proposed EBCLR and independent contributions.
3. Some details should be polished, such as missing reference at L132 and the writing of Section 3, where some sentences are not fluent to follow.
4. As the theoretical part in Section 3 mainly contains intuitive and very straightforward steps, I would recommend the authors to put some more efforts to backup the claim of contributions in following sections.

---

> ### Author Response · Authors · 2022-08-01
> **Response to Reviewer VTiU (Part 3/3)**
>
> **Q5. As the theoretical part in Section 3 mainly contains intuitive and very straightforward steps, I would recommend the authors to put some more efforts to back up the claim of contributions in following sections.**
>
> A. As mentioned in the answers to **Q2** and **Q3**, our experimental contributions are already backed up by thorough experiments and ablation studies. We have also conducted additional evaluations; in Appendix E.4, we have repeated the main experiments with K-Nearest Neighbor classifier in place of linear classifier to cross check the performance of EBCLR, and observed that the conclusions do not change. We also believe the intuitive and straightforward nature of our theory behind EBCLR to be an advantage, rather than a drawback, as EBCLR can appeal to a wider range of machine learning practitioners.
>
> Just in case the Reviewer questions the theoretical soundness of EBCLR, we also provide another theoretical justification on how solving the EBCLR objective Eq. (11) with the approximations of Section 3.2 (and $\lambda = 1$) causes $q_\theta(v,v′)$ to approximate $p(v,v’)$. The following content has been added to Appendix C.2 of our revised paper.
>
> It is known that optimizing the first term of Eq. (11) with Eq. (18) in place of $q_\theta(v’ \mid v)$ will cause Eq. (18) to approximate $p(v’ \mid v)$ [5]. Also, optimizing the second term of Eq. (11) with Eq. (15) or (17) in place of $q_\theta(v)$ will cause Eq. (15) or (17) to be proportional to $p(v)$ (indeed, in Appendix E.5 of the revised paper, we see SGLD samples of the EBCLR marginal $q_\theta(v)$ approximated by Eq. (17) achieve a non-trivial FID score). Thus, the product of Eq. (15) or (17) with Eq. (18) will be proportional to $p(v,v’)$. Moreover, by construction, the product of Eq. (15) or (17) with Eq. (18) is (approximately) $q_\theta(v,v’)$. Hence, optimizing the EBCLR objective Eq. (11) will cause $q_\theta(v,v’)$ to model $p(v,v’)$.
>
> For convenience of the Reviewer, we rewrite the relevant equations below.
>
> Eq. (11): $\mathcal{L}(\theta) \coloneqq \mathbf{E}_p[\log q_\theta(v’ \mid v)] + \lambda \mathbf{E}_p[\log q_\theta(v)]$
>
> Eq. (15): $\widehat q_\theta(v_n) = \frac{1}{N'} \sum_{v_m' : v'_m \neq v_n} q_\theta(v_n, v'_m)$
>
> Eq. (17): $\widehat q_\theta(v_n) = \frac{1}{N} \sum_{m = 1}^N q_\theta(v_n, v'_m)$
>
> Eq. (18): $q_\theta(v_n' \mid v_n) \simeq \frac{q_\theta(v_n, v'_n)}{\widehat q_\theta(v_n)} = \frac{q_\theta(v_n,v'_n)}{\frac{1}{N'} \sum q_\theta(v_n,v'_m)} = \frac{e^{-\|\| z_n - z'_n \|\|^2 / \tau}}{\frac{1}{N'} \sum e^{-\|\| z_n - z'_m \|\|^2 / \tau}}$ where the sums are over $v_m' : v'_m\neq v_n$
>
> **References**
>
> [5] Understanding hard negatives in noise contrastive estimation, NAACL, 2021.

---

> ### Author Response · Authors · 2022-08-01
> **Response to Reviewer VTiU (Part 2/3)**
>
> **Q1 (Continued). Relation of our work to [1,2].**
>
> *Relation to [2]*
>
> [2] proposes to training EBM with an autoencoder structure by minimizing the sum of an encoding loss and a decoding loss to learn an energy surface. A byproduct is that the EBM learns a compressed representation of the data, which can be used for, e.g., denoising. Our work is similar in the aspect that we use a particular EBM architecture (a triplet network) and training the EBM with our proposed method results in a compressed representation of the data which is useful for downstream tasks, e.g., classification and transfer learning. In that sense, our work is also roughly related to Denoising Autoencoders (DAEs) [4] and Restricted Boltzmann Machines (RBMs) as well. Both are EBMs whose goal is to extract useful representations from data. However, we cannot say RBMs, DAEs, or [2] precede our work in relating contrastive learning to EBMs, because they do not mention anything about contrasting positive pairs against negative pairs.
>
> **Q2. Some parts of the paper are not well organized. For example, at the end of the Introduction section, the authors claim five Contributions, but not all of them are well organized in the experiment section and following discussions.**
>
> A. We inform the Reviewer that each of our five contributions (in the paper before revision) corresponds to one section / sub-section of our paper, as written below. In the revised version of our paper, we denoted the relevant section next to each contribution to improve organization.
>
> * “A novel contrastive learning method called EBCLR is proposed by learning the joint distribution of positive pairs. We show that EBCLR loss is equivalent to a combination of a contrastive term and a generative term. To the best of our knowledge, this is the first work to apply EBMs to contrastive learning of visual representations.” &rarr; Section 3
>
> * “To accelerate the training of EBCLR, we show that it is necessary to use an appropriate variance schedule in SGLD” &rarr; Section 4.4
>
> * “By exploring the sensitivity of EBCLR to the generative term, we propose a further improvement by appropriately weighing the generative term” &rarr; Section 4.3
>
> * “EBCLR is several times more sample-efficient than contrastive and non-contrastive methods.” &rarr; Section 4.1
>
> * “Unlike SimCLR, EBCLR is shown to be robust to small number of negative pairs” &rarr; Section 4.2
>
> **Q3. Besides this, it would be not convincing to put some minor points about of the proposed EBCLR and independent contributions.**
>
> A. Correct us if we are wrong, but our understanding of this comment is that some contributions are too minor to be considered as independent contributions. Accordingly, in the revised paper, we re-organized contributions into the following three points by grouping minor contributions.
>
> * We propose a novel contrastive learning method called EBCLR which learns the joint distribution of positive pairs. We show that EBCLR loss is equivalent to a combination of a contrastive term and a generative term (Section 3.) To the best of our knowledge, this is the first work to apply EBMs to contrastive learning of visual representations.
>
> * We show that EBCLR offers two advantages over conventional contrastive learning methods: EBCLR is several times more sample efficient (Section 4.1) and robust to small batch sizes (Section 4.2). These factors lead to a non-trivial performance gain for EBCLR.
>
> * We perform thorough ablation studies of the components of EBCLR: effect of changing the weight of the generative term (Section 4.3), effect of projection space dimension (Section 4.3), and the effect of the proposed SGLD modifications (Section 4.4).
>
> **Q4. Some details should be polished, such as missing reference at L132 and the writing of Section 3, where some sentences are not fluent to follow.**
>
> A. We fixed the missing reference at L132. We thoroughly proof-read the entire paper and fixed grammar mistakes and typos. Also, in Section 3, we added some details to help understand EBCLR. It would be most helpful if the Reviewer pointed out which sentences were unclear, for then we can directly address the Reviewer’s concern.

---

> > ### Comment · Reviewer_VTiU · 2022-08-07
> > **Addressed my concern in paper writing and delivery**
> >
> > I sincerely appreciate the efforts from the authors in polishing the paper. The added new details and the references of equations and pointers to other parts of the paper make it easier for me to follow the paper flow. Moreover, the contribution claim and paper organization is much clearer now. My concerns here have been well addressed.

---

> ### Author Response · Authors · 2022-08-01
> **Response to Reviewer VTiU (Part 1/3)**
>
> **Q1. Relation of our work to [1,2].**
>
> A. We indeed found our work to be relevant to [1,2]. We thank the Reviewer for providing missing references. Below, we explain how our work is similar, and different from [1,2]. We also provide additional explanations that could help the Reviewer understand our work better. (After revision, [1,2] are now included in the Related Works section. An even more extensive survey and discussion of related works has been included in Appendix A of our revised paper.)
>
> *Relation to contrastive divergence*
>
> Contrastive divergence, initially proposed in [3], is a particular method for solving the maximum-likelihood estimation problem for EBMs. Intuitively, it proceeds by minimizing the energy function on data and maximizing the energy function on (possibly short-run) MCMC samples of the current EBM distribution through gradient ascent. In fact, we use an instance of contrastive divergence to optimize the generative term in our EBCLR objective Eq. (11).
>
> We would also like to disambiguate contrastive learning from contrastive divergence. They are similar in the sense that they contrast certain pairs of data. Contrastive learning contrasts positive pairs and negative pairs. Contrastive divergence contrasts data and MCMC samples. However, they differ in the aspect that contrastive learning is inherently a discriminative learning method (in contrastive learning, the model “predicts” the positive pair among negative pairs) and contrastive divergence is a generative learning method. In the context of unsupervised visual representation learning, to the best of our knowledge, there are no works which use discriminative and generative losses simultaneously. Our work, for the first time, explores the synergy between contrastive learning (the discriminative term in Eq. (11)) and contrastive divergence (the generative term in Eq. (11)) for learning visual representations.
>
> *Relation to the Contrastive Energy-Based Model (CEM) [1]*
>
> According to the terms introduced in [1], our EBCLR model distribution Eq. (6) with $\tau=1$ is identical to the model distribution of Non-Parametric CEM (Eq. (4) in [1]). This is because the squared L2 norm between two unit-norm vectors $z$, $z’$ is equivalent to the inner product between $z$ and $z’$ up to an additive constant. However, our work differs from [1] in three aspects.
>
> * Objective function: [1] optimizes the log-likelihood of the joint model distribution $q_\theta(v,v’)$ directly whereas we use Bayes’ rule to decompose the objective into two terms, the discriminative term and the generative term. This allows us to reweight the generative term to learn better visual representations (as explored in Section 4.3).
>
> * Optimization method: [1] maximizes the log-likelihood of the joint model distribution directly via adversarial training. On the other hand, we use gradient ascent to optimize the discriminative term and contrastive divergence to optimize the generative term. Computation costs are similar, as adversarial training also requires inner iterations to generate adversarial examples at every model update step.
>
> * Application of interest: [1] focuses on enhancing image generation, while we focus on learning visual representations useful for downstream tasks.
>
> [1] also establishes a connection between contrastive divergence and InfoNCE (Section 5.1 in [1]). Specifically, the authors of [1] show that when we approximate the model marginal $q_\theta(v)$ with the data distribution $p(v)$, MLE gradient for $q_\theta(v,v’)$ becomes the InfoNCE loss gradient. Surprisingly, we find this holds in our framework as well. If we set $q_\theta(v) = p(v)$ in Eq. (11), the generative term in Eq. (11) becomes independent of $\theta$, so we end up optimizing only the discriminative term. We have shown that the discriminative term can be approximated by a contrastive learning objective, so we recover the result of [1]. In this sense, the derivations in Section 3 of our paper generalizes the connection discovered in [1], as we work with weaker assumptions (we do not set $q_\theta(v) = p(v)$ in Eq. (11)).
>
> **References**
>
> [1] A Unified Contrastive Energy-based Model for Understanding the Generative Ability of Adversarial Training, ICLR, 2022
>
> [2] A Unified Energy-Based Framework for Unsupervised Learning, AISTATS, 2007.
>
> [3] Training Products of Experts by Minimizing Contrastive Divergence. Neural Computation, 2002.
>
> [4] A connection between score matching and denoising autoencoders, Neural Computation, 2011.

---

> > ### Comment · Reviewer_VTiU · 2022-08-07
> > **The connection between contrastive and generative learning**
> >
> > I sincerely thank the authors for the detailed response and the revised draft, which is after a heavy polishing and the current quality is much better.
> >
> > For the connection between contrastive and generative learning methods, the claim **"In the context of unsupervised visual representation learning, to the best of our knowledge, there are no works which use discriminative and generative losses simultaneously"** might be not that accurate. It is true that typical representation learning usually uses only contrastive or generative models. But combining these two sources for training has been shown in many downstream works, such as [1].
> >
> > Moreover, on the theoretical side, even limited in the contrastive SSL with negative samples, there are some previous methods have looked into bridging generative and discriminative methods, such as [2], and also [3] which it is based on. I notice that you don't put that claim as the main contribution in the Introduction, which I think is proper. And though it may not be necessary, it might be good to provide more analytical study about the role of negative samples in bridging (1) EMB and contrastive and (2) discriminative and generative SSL methods, for which [3] might be able to provide some information.
> >
> > Referece.
> >
> > [1] Chen H, Wang Y, Lagadec B, et al. Joint generative and contrastive learning for unsupervised person re-identification[C]//Proceedings of the IEEE/CVF conference on computer vision and pattern recognition. 2021: 2004-2013.
> >
> > [2] Zimmermann, Roland S., et al. "Contrastive learning inverts the data generating process." International Conference on Machine Learning. PMLR, 2021.
> >
> > [3] Wang, Tongzhou, and Phillip Isola. "Understanding contrastive representation learning through alignment and uniformity on the hypersphere." International Conference on Machine Learning. PMLR, 2020.

---

> > > ### Author Response · Authors · 2022-08-08
> > > **Response to "The connection between contrastive and generative learning"**
> > >
> > > **Q1. For the connection between contrastive and generative learning methods, the claim *"In the context of unsupervised visual representation learning, to the best of our knowledge, there are no works which use discriminative and generative losses simultaneously"* might be not that accurate. It is true that typical representation learning usually uses only contrastive or generative models. But combining these two sources for training has been shown in many downstream works, such as [1].**
> > >
> > > **Moreover, on the theoretical side, even limited in the contrastive SSL with negative samples, there are some previous methods have looked into bridging generative and discriminative methods, such as [2], and also [3] which it is based on. I notice that you don't put that claim as the main contribution in the Introduction, which I think is proper.**
> > >
> > > A. We acknowledge the Reviewer’s comment. Hence, we have removed the sentence of concern from Appendix A.1 and added the references [1,2,3]. Specifically, in Appendix A.1, we have revised the content as follows:
> > >
> > > *Before Revision*
> > >
> > > In the context of unsupervised visual representation learning, to the best of our knowledge, there are no works which use discriminative and generative losses simultaneously. Our work, for the first time, explores the synergy between contrastive learning (the discriminative term in Eq. (11)) and contrastive divergence (the generative term in Eq. (11)) for learning visual representations.
> > >
> > > *After Revision*
> > >
> > > In the context of unsupervised visual representation learning, there are several works which connect contrastive learning with generative learning [1,2,3]. For instance, Wang et al. [1] combine contrastive learning with GANs to improve person re-identification. Zimmermann et al. [2] and Wang et al. [3] study the distributional properties of embeddings found by contrastive learning. Our work, for the first time, explores the synergy between contrastive learning (the discriminative term in Eq. (11)) and EBMs / contrastive divergence (the generative term in Eq. (11)) for learning visual representations.
> > >
> > > **Q2. And though it may not be necessary, it might be good to provide more analytical study about the role of negative samples in bridging (1) EMB and contrastive and (2) discriminative and generative SSL methods, for which [3] might be able to provide some information.**
> > >
> > > A. We agree with the Reviewer that a theoretical study of the role of negative samples in our framework is a non-trivial and interesting topic. In our framework, negative samples serve as points used in Monte-Carlo estimates of the model marginal distributions Eq. (15) & Eq. (17). Hence, we expect increasing the number of negative samples would reduce the variance of estimates Eq. (15) & Eq. (17). Empirical results in Section 4.2 suggest that EBCLR is robust to variance in Eq. (15) & Eq. (17), as linear evaluation accuracy is almost invariant to the number of negative samples. We believe that a precise theoretical characterization of the effect of this variance on EBCLR would be a meaningful research topic of its own, so we leave it for future work.
> > >
> > > We have also taken some time to read the recommended paper [3] and found the proposed ideas to be intriguing. However, we believe it will take more than just a few days to derive meaningful analytical results that connect [3] to EBCLR and that the results would be worthy of a paper of its own. Hence, we must ask for your kind understanding if we are unable to provide more analytical study about the role of negative samples within the remaining two days of the author-reviewer discussion period.
> > >
> > > [1] Chen H, Wang Y, Lagadec B, et al. Joint generative and contrastive learning for unsupervised person re-identification[C]//Proceedings of the IEEE/CVF conference on computer vision and pattern recognition. 2021: 2004-2013.
> > >
> > > [2] Zimmermann, Roland S., et al. "Contrastive learning inverts the data generating process." International Conference on Machine Learning. PMLR, 2021.
> > >
> > > [3] Wang, Tongzhou, and Phillip Isola. "Understanding contrastive representation learning through alignment and uniformity on the hypersphere." International Conference on Machine Learning. PMLR, 2020.

---

> ### Author Response · Authors · 2022-08-07
> **Does our response answer your concerns and questions?**
>
> Dear Reviewer VTiU,
>
> As the end of the author-reviewer discussion period is approaching, we would like to ask whether our response has addressed your concerns and questions adequately. If not, we would be happy to discuss further.
>
> Regards,
>
> the authors.

---

> > ### Comment · Reviewer_VTiU · 2022-08-07
> > **Response to the author rebuttal.**
> >
> > Thanks for the rebuttal and the well-revised paper draft. Sorry for the late reply as I spent some time to re-read some related works in this area to get more accurate calibration of this paper's contribution and novelty.
> >
> > Most of my concerns have been addressed and I think the current version better delivers an accurate message to the readers. I have some minor concerns about the novelty of connecting generative and discriminative methods in unsupervised learning. But considering this is not the main contribution claimed for this paper, I am comfortable with the current paper delivery. I recognize the contribution of combing EMBs and Contrastive learning and the thorough experiments to backup the performance of the method.
> >
> > I would raise my score and continue to seek more sense of this paper in the coming discussion stages.

---

> > > ### Author Response · Authors · 2022-08-08
> > > **Thank you!**
> > >
> > > Thank you for the reconsideration of our paper and raising the score! Your comments have helped us improve the presentation of our paper and better relate our work to previous studies. Regarding the minor concerns, we have provided additional feedback in the reply to your comment “The connection between contrastive and generative learning”.

---

### Official Review · Reviewer_wreL · 2022-07-16

**Rating:** 7
**Confidence:** 3
**Soundness:** 3 good
**Presentation:** 3 good
**Contribution:** 4 excellent

**Summary:**

The paper proposes a representations learning method, Energy-Based Contrastive Learning (EBCLR), that combines contrastive learning with energy-based models (EBMs). The authors not only provide theoretical formulation of the proposed framework but also back up all the design choices empirically by providing adequate ablations.

The efficiency of the method is demonstrated on MNIST, CIFAR-10 and CIFAR-100 where it outperforms contrastive learning baselines in sample efficiency and linear evaluations.

**Questions:**

- How do you initialise SGLD in the very first step when there are no previous iterations for generated samples (or replay buffer is empty)?
- I know the paper focuses on EBCLR’s representation learning capability, however, since there is a generative loss in the objective function, what’s the generative performance of the model? Does contrastive learning help it or deteriorate it?
- Given the scaling difficulties associated with EBMs, could the contrastive learning be paired with other generative loss to a similar effect?

**Limitations:**

The main limitations seem to be around experimental evaluation. I would be happy to increase my score if authors could demonstrate the benefits on one of the large scale dataset.

There doesn't seem to be any obvious negative societal impact.

**Strengths And Weaknesses:**

**Strengths:**
- The paper proposes a simple extension of contrastive learning frameworks to energy based models for learning better representations efficiently.
- The proposed method outperforms contrastive baselines and reach convergence (4x to 20x) faster than the baselines.
- The authors provide theoretical formulation of their newly proposed framework.
- Detailed ablations are provided to justify all the design choices.

**Weaknesses:**
- Currently, the related work section is more about explaining the inner workings of contrastive and energy based methods instead of focusing on the prior or similar works. Although I am not highly familiar with literature on energy based models but some insights from contrastive divergence in energy based models [2] (or differences with the present work [1]) would go a long way in helping new readers.
- I would appreciate if the authors could position their work and contributions appropriately w.r.t to the prior work [1].
- Although it's not a weakness, but I would have appreciated if the authors demonstrated benefits of EBCLR on one of the large-scale datasets.

**Minor Comments:**
- In abstract, you used terms positive and negative pairs without introducing them. It’s okay for the readers who know about contrastive learning, but someone who is new to the domain wouldn’t get it.
- Missing appendix reference in line 132.
- Could you please add subscript $\textit(t)$ in Eq. 21 similar to update steps in Sec. 2.2, Eq. 5?
- Line 193, repetition of $\pi_{\theta}$.

[1] https://openreview.net/forum?id=XhF2VOMRHS
[2] http://yann.lecun.com/exdb/publis/pdf/ranzato-unsup-07.pdf

---

> ### Author Response · Authors · 2022-08-01
> **Response to Reviewer wreL (Part 3/3)**
>
> **Q9. Given the scaling difficulties associated with EBMs, could the contrastive learning be paired with other generative loss to a similar effect?**
>
> A. The following discussion has been added to Appendix F of our revised paper.
>
> Since the main idea of our method is to use the joint distribution of positive pairs as a measure of semantic similarity of images, one would necessarily need to match the model distribution $q_\theta(v,v’)$ to the distribution of positive pairs $p(v,v’)$ if we want to achieve a similar effect. If we solve the decomposition Eq. (11), we would need to match $q_\theta(v)$ to $p(v)$. We could consider three alternative losses / methods for doing this: noise contrastive estimation (NCE) [13], VERA [14], and score matching [15]. All three are methods for learning densities only with samples from the target distribution ($p(v,v’)$ or $p(v)$ in this case).
>
> Although the NCE loss does not depend on MCMC, it requires specification of a noise distribution whose normalized density can be easily evaluated. In most cases, such noise distribution samples are easily distinguishable from natural image samples and lead to the density chasm problem [16], making optimization difficult. Telescoping density-ratio estimation [16] attempts to mitigate this problem, but it does not seem to scale well to large data. Flow contrastive estimation [17] uses a flow model to define the noise distribution and trains a flow model along with the EBM. Though flow contrastive estimation requires an auxiliary flow model, we believe this is most promising alternative to MCMC. In a similar sense, VERA [14], which uses a generator to sample from the EBM, seems promising as well. Yet, flow contrastive estimation and VERA also did not demonstrate scalability to large-scale datasets such as ImageNet. Score matching matches the gradient of model log-density to the gradient of data log-density, so it does not rely on MCMC. However, it requires calculation of the Hessian, so in the deep learning setting, this approach is also quite expensive.
>
> **References**
>
> [13] Noise-contrastive estimation: A new estimation principle for unnormalized statistical models, AISTATS, 2010.
>
> [14] No MCMC for me: Amortized sampling for fast and stable training of energy-based models, ICLR, 2021
>
> [15] Estimation of Non-Normalized Statistical Models by Score Matching, JMLR, 2005.
>
> [16] Telescoping density-ratio estimation, NeurIPS, 2020.
>
> [17] Flow Contrastive Estimation of Energy-Based Models, CVPR, 2020..

---

> ### Author Response · Authors · 2022-08-01
> **Response to Reviewer wreL (Part 2/3)**
>
> **Q1 (Continued). Relation of our work to [1,2].**
>
> *Relation to [2]*
>
> [2] proposes to training EBM with an autoencoder structure by minimizing the sum of an encoding loss and a decoding loss to learn an energy surface. A byproduct is that the EBM learns a compressed representation of the data, which can be used for, e.g., denoising. Our work is similar in the aspect that we use a particular EBM architecture (a triplet network) and training the EBM with our proposed method results in a compressed representation of the data which is useful for downstream tasks, e.g., classification and transfer learning. In that sense, our work is also roughly related to Denoising Autoencoders (DAEs) [4] and Restricted Boltzmann Machines (RBMs). Both are EBMs whose goal is to extract useful representations from data. However, we cannot say RBMs, DAEs, or [2] precede our work in connecting contrastive learning to EBMs, because they do not mention anything about contrasting positive pairs against negative pairs.
>
> **Q2. In abstract, you used terms positive and negative pairs without introducing them. It’s okay for the readers who know about contrastive learning, but one who is new to the domain wouldn’t get it.**
>
> A. We updated our paper to explain the terms positive and negative pairs in the abstract.
>
> **Q3. Missing appendix reference in line 132.**
>
> A. We fixed the missing appendix reference.
>
> **Q4. Add subscript t in Eq. 21 similar to update steps in Sec. 2.2, Eq. 5.**
>
> A. We added the subscript t.
>
> **Q5. Line 193, repetition of pi_\theta.**
>
> A. We removed the redundant pi_\theta.
>
> **Q6. Although it's not a weakness, but I would have appreciated if the authors demonstrated benefits of EBCLR on one of the large-scale datasets.**
>
> A. We tried to run EBCLR on ImageNet. But (for reasons outlined in the limitations section / Section 5 of this paper) we were unable to produce results within the rebuttal period as we had to run hyperparameter searches for the baselines and EBCLR. We also note that many published works on EBMs [5-8] and adversarial training [9-12] (which often requires inner iterations to generate adversarial examples at every model update step) train models only up to medium-scale datasets such as CIFAR-10/100. If we obtain ImageNet results, we will make sure to update our paper.
>
> **Q7. How do you initialize SGLD in the very first step when there are no previous iterations for generated samples (or replay buffer is empty)?**
>
> A. At the beginning of training, we fill the buffer entirely with proposal distribution samples. This detail is added to Appendix D of our revised paper.
>
> **Q8. Effect of contrastive loss on the generative performance of EBCLR.**
>
> A. The following discussion has been added to Appendix E.5 of our revised paper.
>
> To address this question, we optimized
>
> $\max_\theta \gamma \mathbb{E}_p [\log q_\theta(v' \mid v)] + \mathbb{E}_p[\log q_\theta(v)]$
>
> for various values of $\gamma$ on FMNIST. Note that in the original EBCLR objective, weight $\lambda$ comes in front of the generative term. We then measured the Fréchet Inception Distance (FID) between samples from the true marginal $p(v)$ and SGLD samples from the EBM marginal $q_\theta(v)$ approximated by Eq. (19) / (17). The results are shown below.
>
> | $\gamma$ | 0 | 0.01 | 0.1 | 1.0 | 10.0 |
> | --- | --- | --- | --- | --- | --- |
> |FID $\downarrow$ | 40.78 | 9.68 | 7.68 | 17.03 | 139.19 |
>
> Interestingly, we found using an appropriate $\gamma > 0$ led to improvements in FID over $\gamma = 0$. That is, contrastive learning did help improve the generative performance of the EBM. However, using an excessively large $\gamma$ led to a deterioration of the performance. This is analogous to the result of Section 4.3 where we observed using an appropriate weight on the generative term led to improvements over using only the discriminative term (contrastive loss). The difference is that to achieve optimal generative performance, the generative term weight needs to be larger (i.e., $\gamma < 1$) whereas to achieve optimal discriminative performance, the discriminative term weight needs to be larger (i.e., $\lambda < 1$ in Eq. (11)). We speculate using a larger model could mitigate this trade-off, but this topic is beyond the scope of this work.
>
> **References**
>
> [5] Your classifier is secretly and energy-based model and you should treat it like one, ICLR, 2020.
>
> [6] Learning the Stein Discrepancy for Training and Evaluating Energy-Based Models without Sampling, ICML, 2020
>
> [7] Flow Contrastive Estimation of Energy-Based Models, CVPR, 2020.
>
> [8] JEM++: Improved Techniques for Training JEM, ICCV, 2021.
>
> [9] Theoretically Principled Trade-off between Robustness and Accuracy, ICML, 2019.
>
> [10] Unlabeled Data Improves Adversarial Robustness, NeurIPS, 2019.
>
> [11] Improving Adversarial Robustness Requires Revisiting Misclassified Examples, ICLR, 2020.
>
> [12] Learning Adversarially Robust Representations via Worst-Case Mutual Information Maximization, ICML, 2020.

---

> ### Author Response · Authors · 2022-08-01
> **Response to Reviewer wreL (Part 1/3)**
>
> **Q1. Relation of our work to [1,2].**
>
> A. We indeed found our work to be relevant to [1,2]. We thank the Reviewer for providing missing references. Below, we explain how our work is similar, and different from [1,2]. We also provide additional explanations that could help the Reviewer understand our work better. (After revision, [1,2] are now included in the Related Works section. An even more extensive survey and discussion of related works has been included in Appendix A of our revised paper.)
>
> *Relation to contrastive divergence*
>
> Contrastive divergence, initially proposed in [3], is a particular method for solving the maximum-likelihood estimation problem for EBMs. Intuitively, it proceeds by minimizing the energy function on data and maximizing the energy function on (possibly short-run) MCMC samples of the current EBM distribution through gradient ascent. In fact, we use an instance of contrastive divergence to optimize the generative term in our EBCLR objective Eq. (11).
>
> We would also like to disambiguate contrastive learning from contrastive divergence. They are similar in the sense that they contrast certain pairs of data. Contrastive learning contrasts positive pairs and negative pairs. Contrastive divergence contrasts data and MCMC samples. However, they differ in the aspect that contrastive learning is inherently a discriminative learning method (in contrastive learning, the model “predicts” the positive pair among negative pairs) and contrastive divergence is a generative learning method. ~~In the context of unsupervised visual representation learning, to the best of our knowledge, there are no works which use discriminative and generative losses simultaneously.~~ ** Our work, for the first time, explores the synergy between contrastive learning (the discriminative term in Eq. (11)) and contrastive divergence (the generative term in Eq. (11)) for learning visual representations.
>
> ** This sentence was deleted after a discussion with Reviewer VTiU. We refer the Reviewer to the comment https://openreview.net/forum?id=GwwC16ECrM5&noteId=s7jwMq0GVZh for more details.
>
> *Relation to the Contrastive Energy-Based Model (CEM) [1]*
>
> According to the terms introduced in [1], our EBCLR model distribution Eq. (6) with $\tau=1$ is identical to the model distribution of Non-Parametric CEM (Eq. (4) in [1]). This is because the squared L2 norm between two unit-norm vectors $z$, $z’$ is equivalent to the inner product between $z$ and $z’$ up to an additive constant. However, our work differs from [1] in three aspects.
>
> * Objective function: [1] optimizes the log-likelihood of the joint model distribution $q_\theta(v,v’)$ directly whereas we use Bayes’ rule to decompose the objective into two terms, the discriminative term and the generative term. This allows us to reweight the generative term to learn better visual representations (as explored in Section 4.3).
>
> * Optimization method: [1] maximizes the log-likelihood of the joint model distribution directly via adversarial training. On the other hand, we use gradient ascent to optimize the discriminative term and contrastive divergence to optimize the generative term. Computation costs are similar, as adversarial training also requires inner iterations to generate adversarial examples at every model update step.
>
> * Application of interest: [1] focuses on enhancing image generation, while we focus on learning visual representations useful for downstream tasks.
>
> [1] also establishes a connection between contrastive divergence and InfoNCE (Section 5.1 in [1]). Specifically, the authors of [1] show that when we approximate the model marginal $q_\theta(v)$ with the data distribution $p(v)$, MLE gradient for $q_\theta(v,v’)$ becomes the InfoNCE loss gradient. Surprisingly, we find this holds in our framework as well. If we set $q_\theta(v) = p(v)$ in Eq. (11), the generative term in Eq. (11) becomes independent of $\theta$, so we end up optimizing only the discriminative term. We have shown that the discriminative term can be approximated by a contrastive learning objective, so we recover the result of [1]. In this sense, the derivations in Section 3 of our paper generalizes the connection discovered in [1], as we work with weaker assumptions (we do not set $q_\theta(v) = p(v)$ in Eq. (11)).
>
> **References**
>
> [1] A Unified Contrastive Energy-based Model for Understanding the Generative Ability of Adversarial Training, ICLR, 2022
>
> [2] A Unified Energy-Based Framework for Unsupervised Learning, AISTATS, 2007.
>
> [3] Training Products of Experts by Minimizing Contrastive Divergence. Neural Computation, 2002.
>
> [4] A connection between score matching and denoising autoencoders, Neural Computation, 2011.

---

> ### Author Response · Authors · 2022-08-07
> **Does our response answer your concerns and questions?**
>
> Dear Reviewer wreL,
>
> As the end of the author-reviewer discussion period is approaching, we would like to ask whether our response has addressed your concerns and questions adequately. If not, we would be happy to discuss further.
>
> Regards,
>
> the authors.

---

### Author Response · Authors · 2022-08-01
**General Reply to All Reviewers**

We sincerely thank all the Reviewers for the valuable comments. We have tried our best to incorporate all suggestions, comments, and questions into the revised version of our paper. The updated parts in the paper are highlighted in blue. Below, we list the changes in the revised version.

*Appendix is now included in the main submission.*

**Abstract and Introduction** We removed or reworded some possibly misleading sentences and organized the contributions section.

**Related Works** We added missing references, including ones pointed out by Reviewers wreL and VTiU.

**Figure 1 (left)** We simplified some notations to improve clarity.

**Section 3** We added more explanation of the rationale behind EBCLR.

**Appendix A (Additional discussion)** More extensive discussion of related works on EBMs, contrastive divergence, contrastive learning, and generative models.

**Appendix C.2 (Additional theory)** Theoretical justification that EBCLR works as intended even with the approximations in Section 3.2.

**Appendix E.4 (Additional experiment)** Evaluation with KNN classifiers.

**Appendix E.5 (Additional experiment)** Effect of contrastive loss on the generative performance of EBCLR.

**Appendix F (Additional discussion)** A discussion on making EBCLR scalable by using other generative losses.

---

### Meta-Review · Area_Chair_LuFv · 2022-08-29

**Recommendation:** Accept
**Confidence:** Certain

**Metareview:**

The paper connects contrastive learning and energy-based models and proposes a new variant of contrastive learning based on SGLD. All of the reviewers believe the paper is a good fit for NeurIPS, and I recommend acceptance. That said, as reviewers point out, results on ImageNet are also expected.

**Award:**

No

---

### Decision · Program_Chairs · 2022-09-14

Accept